# PIXAB-CAM: ATTEND PIXEL, NOT CHANNEL

## ABSTRACT

To understand the internal behaviors of convolution neural networks (CNNs), many class activation mapping (CAM) based methods, which generate an explanation map by a linear combination of channels and corresponding weights, have been proposed. Previous CAM-based methods have tried to define a channel-wise weight that represents the importance of a channel for the target class. However, these methods have two common limitations. First, all pixels in the channel share a single scalar value. If the pixels are tied to a specific value, some of them are overestimated. Second, since the explanation map is the result of a linear combination of channels in the activation tensor, it is inevitably dependent on the activation tensor. To address these issues, we propose gradient-free Pixel-wise Ablation-CAM (Pixab-CAM), which utilizes pixel-wise weights rather than channel-wise weights to break the link between pixels in a channel. In addition, in order not to generate an explanation map dependent on the activation tensor, the explanation map is generated only with pixel-wise weights without linear combination with the activation tensor. In this paper, we also propose novel evaluation metrics to measure the quality of explanation maps using an adversarial attack. We demonstrate through experiments the qualitative and quantitative superiority of Pixab-CAM.

## 1 INTRODUCTION

As the performance of convolution neural networks (CNNs) converges to some extent(Szegedy et al., 2015; Simonyan & Zisserman, 2014; Huang et al., 2017; Howard et al., 2017; Chollet, 2017), researchers have become interested in interpreting their predictions(Springenberg et al., 2014; Smilkov et al., 2017; Fong & Vedaldi, 2017; Bach et al., 2015; Lundberg & Lee, 2017; Ribeiro et al., 2016; Petsiuk et al., 2018; Shrikumar et al., 2017). One of them is class activation mapping (CAM) based approach(Zhou et al., 2016), which generates an explanation map by a linear combination of channels in the activation tensor and corresponding weights. The explanation map $L_{ij}^c$ for the target class $c$ is calculated as Eqn.1. $A_{ij}^k$ represents activation of pixel at spatial location $(i, j)$ for $k$-th channel of the activation tensor $A$, and $w_k^c$ denotes the coefficient of the $k$-th channel for class $c$.

$$L_{ij}^c = \sum_k w_k^c \cdot A_{ij}^k \qquad (1)$$

The activation tensor $A$ is fixed since it is determined by the given input image and pretrained model(LeCun et al., 1998). Therefore, $L^c$ depends on how $w_k^c$ is defined. Thus, different $w_k^c$ have been proposed in many CAM-based methods to generate high-quality explanation maps. The previous CAM-based methods can be divided into two categories, one is gradient-based CAMs(Selvaraju et al., 2017; Chattopadhay et al., 2018; Fu et al., 2020), and the other is gradient-free CAMs(Wang et al., 2020; Ramaswamy et al., 2020). Gradient-based CAMs and gradient-free CAMs can be distinguished by whether or not gradients are used in the process of deriving weights.

### 1.1 GRADIENT-BASED CAMS

Grad-CAM(Selvaraju et al., 2017) obtains the weight of $k$-th channel by averaging their gradients. However, it has a problem in that the weight is crushed due to the offset of positive and negative gradients. ($Z$ is the number of pixels in a channel and $Y^c$ is the classification score for class $c$)

$$w_k^c = \frac{1}{Z} \sum_i \sum_j \frac{\partial Y^c}{\partial A_{ij}^k} \qquad (2)$$

Grad-CAM++(Chattopadhay et al., 2018) introduced higher-order derivatives $\alpha_{ij}^{kc}$ in Grad-CAM equation to solve the problem that channels with fewer footprints fade away in the explanation map.

$$w_k^c = \sum_i \sum_j \alpha_{ij}^{kc} \cdot ReLU \left( \frac{\partial Y^c}{\partial A_{ij}^k} \right) \qquad (3)$$

XGrad-CAM(Fu et al., 2020) points out that previous CAM-based methods lack sufficient theoretical support and propose weight that satisfies two basic axioms: Sensitivity and Conservation.

$$w_k^c = \sum_i \sum_j \left( \frac{A_{ij}^k}{\sum_i \sum_j A_{ij}^k} \frac{\partial Y^c}{\partial A_{ij}^k} \right) \qquad (4)$$

However, Gradient-based CAMs suffer from gradient saturation problems, leading to localization failures for relevant regions.(Wang et al., 2020; Ramaswamy et al., 2020; Jung & Oh, 2021)

## 1.2 GRADIENT-FREE CAMS

Score-CAM(Wang et al., 2020) defines "increase of confidence" as the coefficient of the channel. For input $X$ and baseline input $X_b$, the weight of the $k$-th channel for class $c$ is defined as

$$w_k^c = f(X \circ s(u(A^k))) - f(X_b) \qquad (5)$$

where $u(\cdot)$ indicates the upsampling operation into the input size, $s(\cdot)$ is a function that normalizes all elements to $[0, 1]$ and $\circ$ is the "Hadamard product".

Ablation-CAM(Ramaswamy et al., 2020), which inspired our research, defines the weight of a specific channel as the fraction of drop in activation score of target class when that channel is removed. In the equation below, $Y_k^c$ is the score of class $c$ when the $k$-th channel has been replaced by zero.

$$w_k^c = \frac{Y^c - Y_k^c}{Y^c} \qquad (6)$$

Gradient-free CAMs remove the dependence of gradients, but they are time-comsuming because the prediction process must be repeated as many times as the number of channels to acquire the weights.

Regardless of whether gradient is used or not, all CAM-based methods so far have two things in common. First, they all use channel-wise weights $w_k^c$ (see Eqn.1). However, since the channel-wise weights allow all pixels in a specific channel to share a single weight, it is an unfair weighting method from the pixel perspective. (Although $\alpha_{ij}^{kc}$ in Grad-CAM++ (Eqn.3) modifies the gradients at the pixel level, but these gradients are eventually crushed into a single scalar value, so it doesn't solve the problem of pixels being tied together.) Furthermore, channel-wise weight-based methods tend to assign high weight if there are pixels that have a high contribution to the target class among the active pixels in the channel. Therefore, within a specific channel, other pixels that are active with the pixels related to the target class are also given a high weight. If channels containing these overestimated pixels are accumulated, overestimated pixels also occur in the explanation map.(see Sec.3.1) Second, when generating the explanation map $L^c$, all CAM-based methods follow Eqn.1, which contains the activation tensor $A$. Therefore, the explanation map has no choice but to depend on the activation tensor, and eventually may capture too much meaningless information since the channels in the activation tensor are not necessarily related to the target category(Zhang et al., 2021).(see Sec.3.2)

## 2  PROPOSED EVALUATION METRICS OF THE EXPLANATION MAP

Since it is hard to know where CNN looks and makes a specific prediction, the existing evaluation metrics of the explanation map have limitations. AD & IIC(Chattopadhay et al., 2018) and Deletion & Insertion(Petsiuk et al., 2018) are coarse proxy evaluations because removing pixels by zero-ing/graying is unnatural in real photos(Dabkowski & Gal, 2017). Also, in Localization Evaluation(Wang et al., 2020), there may not be a correlation between localization ability and the quality of the explanation map since the human-annotated bounding box may not match where CNN is actually focusing. However, if we could know where CNN is looking, we could make a more accurate assessment. We use an adversarial attack for this.(we use FGSM attack(Goodfellow et al., 2014).) If a targeted adversarial attack is made on a specific patch in the zero image, it is self-evident that CNN sees that patch and determines that it is a target, so the performance can be evaluated by whether the explanation map correctly captures that part. We propose 2 evaluation tasks (DAP1 and DAP2) from this idea. Before introducing them, we will explain 2 adversarial examples (AE1 and AE2) first.

## 2.1 ADVERSARIAL EXAMPLE 1 (AE1)

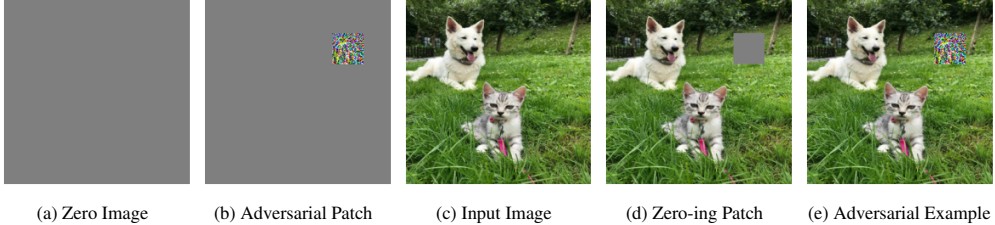

(a) Zero Image     (b) Adversarial Patch     (c) Input Image     (d) Zero-ing Patch     (e) Adversarial Example

Figure 1: Adversarial patch in (b) was attacked to become a "cucumber".

It is simple to attach an adversarial patch classified as a given target class to an input image. First, we apply a targeted adversarial attack so that the patch at a specific location in the zero image is classified into the class we want. Since the attack is applied only to the patch, the rest of the area is still zero as in (b). Then, the input image with only the patch area of zero((d)) is added to the image (b). The finally obtained adversarial example AE1((e)) is used for our evaluation task DAP1.

## 2.2 ADVERSARIAL EXAMPLE 2 (AE2)



(a) Adversarial Patch 1    (b) Adversarial Patch 2    (c) Adversarial Patch 3    (d) Adversarial Patch 4    (e) Adversarial Example

Figure 2: (a): "chihuahua", (b): "tusker", (c): "soccer ball", (d): "broccoli"

For generating a second adversarial example AE2, only adversarial patches are used without input images. This can also be easily created. After generating four adversarial patches in the same way as in Sec.2.1 (b), combine them to make the final adversarial example AE2. (Each adversarial patch is attacked to become a different class.) AE2 created in this way is used for evaluation task DAP2.

## 2.3 DETECTING ADVERSARIAL PATCH

Explanation Map$^{AE1}$    Pointing Game$^{AE1}$    Object Localization$^{AE1}$    Explanation Map$^{AE2}$    Pointing Game$^{AE2}$    Object Localization$^{AE2}$

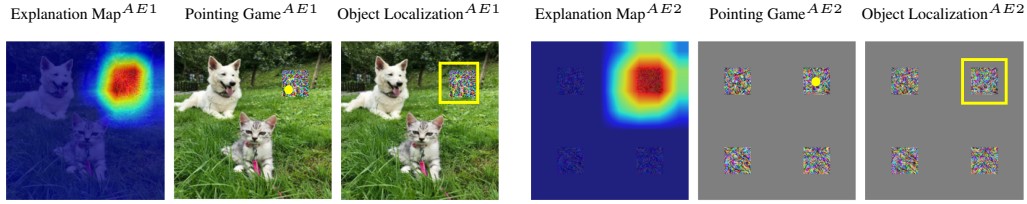

Figure 3: Columns 1-3 and 4-6 correspond to the tasks "Detecting Adversarial Patch 1 (DAP1)" and "Detecting Adversarial Patch 2 (DAP2)". Also, each is an explanation of "cucumber" and "tusker".

In this paper, "Pointing Game" and "Object Localization" are used as metrics in "Detecting Adversarial Patch 1 (DAP1)" and "Detecting Adversarial Patch 2 (DAP2)" tasks.

**Pointing Game (PG) (Petsiuk et al., 2018):** The Pointing Game (PG) is a metric that measures whether the maximum point of the explanation map is inside the human-annotated object bounding box. As in columns 2 and 5 of Figure 3, if the point lies inside the target bounding box, it is counted as a hit. Pointing Game accuracy is calculated as $\frac{\#Hits}{\#Hits+\#Misses}$.

**Object Localization (OL) (Wang et al., 2020):** Object Localization (OL) is a metric that measures how much the energy of the explanation map falls into the target bounding box instead of using only the maximum point. This metric can be formulated as $\frac{\sum L_{ij \in bbox}^{c}}{\sum L_{ij \in bbox}^{c} + \sum L_{ij \notin bbox}^{c}}$.

However, in both DAP1 and DAP2, the ground truth bounding box of the target object is the patch, not the human-annotated bounding box. So, the ground truth bounding box in Sec.5.2.2 is the human-annotated bounding box, and the ground truth bounding box in Sec.5.2.3 is the patch.

# 3 MOTIVATION

## 3.1 CHANNELS WITH OVERESTIMATED PIXELS (COP)

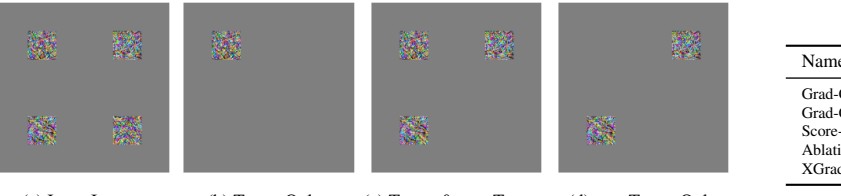

| Name | # COP |
|------|-------|
| Grad-CAM | 56 |
| Grad-CAM++ | 39 |
| Score-CAM | 26 |
| Ablation-CAM | 53 |
| XGrad-CAM | 37 |

(a) Input Image   (b) Target Only   (c) Target & non-Target   (d) non-Target Only

Table 1: # COP

Figure 4: Target: "chihuahua",   non-Target: "tusker", "soccer ball", "broccoli"

In Figure 4, (b)-(d) indicate which part of the input image the specific channels of the activation tensor focus on (Masking input image with upsampled channels.). Channel (b) is focusing only on the target part, channel (c) is focusing on both the target and non-target part and channel (d) is focusing only on the non-target part. We investigated the weights for channels (b)-(d), and it was confirmed that Grad-CAM, Grad-CAM++, Score-CAM, Ablation-CAM, and XGrad-CAM all gave high weights to channels (b) and (c) and low weights to channel (d). In other words, regardless of whether a specific channel contains a non-target part or not, as long as it contains a target part, that channel tends to be given a high weight. However, since channel-wise weight-based methods assign one weight per channel, all pixels in the channel share a single weight. As a result, for a channel like (c), pixels in the non-target part are also given a high weight. As shown in Appendix A, these channels are commonly found in the activation tensor. Since the explanation map of previous methods is generated by a linear combination of channels, if the number of channels with overestimated pixels(COP) increases, the overestimated pixels also appear in the explanation map. These overestimated pixels suppress other pixels, so the explanation map may highlight the wrong places or only part of the target. Table 1 shows the average number of channels focusing on both the target and non-target part among the channels with the top 100 weights(Inception V3 has 2048 channels). (If the channel contains more than 80% of the patch, the channel is considered to be focusing on that patch.) The values in Table 1 are averaged for 500 randomly generated AE2s. Among the five methods, Score-CAM best distinguishes the channel focusing only on the target part and the channel focusing on both the target and non-target part. However, this is also not a satisfactory value.

## 3.2 DEPENDENCY ON ACTIVATION TENSOR (DOA)

**Reason for removing the dependency on activation tensor:**   To check the dependence between the explanation map and the activation tensor, we compare Pixab-CAM, which will be covered in Sec.4, with the previous methods for the "Detecting Adversarial Patch 1 (DAP1)" task presented in Sec.2. In addition, in order to check where the activation values are intensively distributed in the activation tensor, we defined Activation-CAM, which is expressed as $\sum_k 1 \cdot A_{ij}^k$. Since the channel-wise weights are fixed at 1, Activation-CAM can indicate where the activation tensor focuses.

If there is only an adversarial patch in the image, like the first row in Figure 5, all methods detect this patch well. However, if features are evenly distributed in the image, such as when a natural image and an adversarial patch are together (the result of Activation-CAM in the second row of Figure 5), the channel-wise weight-based methods cannot capture the patch properly even if the same patch is in the same location. Since the explanation map is generated from a linear weighted combination of channels in the activation tensor, it is difficult to properly focus on target if the target is located where the activation tensor is sparsely distributed (= the number of channels capturing the target is small), such as the result of Activation-CAM in the second row of the Figure 5 (In this case, the target is a patch.). This situation is common in the real world since channels in the activation tensor are not necessarily related to the target class(Zhang et al., 2021). On the other hand,

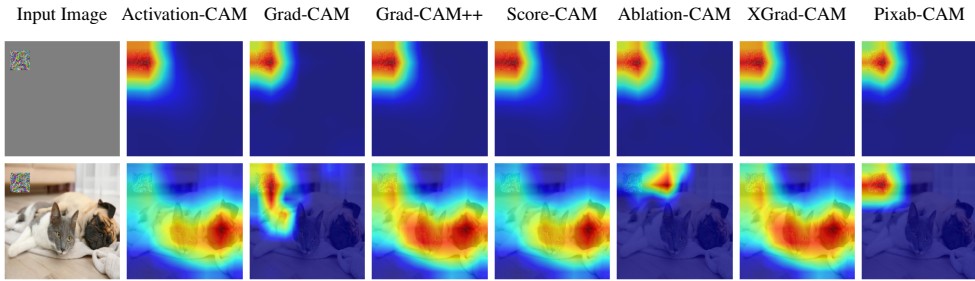

Figure 5: Both the first row and the second row are explanations of "keypad".

Pixab-CAM can handle any situation flexibly regardless of how the activation tensor is distributed on the image because the activation tensor does not participate in the process of generating the explanation map (see Sec.4). Another finding is that the explanation maps generated by channel-wise weight-based methods except Grad-CAM and Ablation-CAM do not deviate much from the result of Activation-CAM, as shown in Figures 5, 6, and 8. In particular, the explanation map of Score-CAM is very similar to the results of Activation-CAM. From this, it can be inferred that the reason that all the channel-wise weight-based methods in the first row of Figure 5 produced the correct explanation map is that the features are distributed only in the adversarial patch, as shown in the result of Activation-CAM. In other words, since there were no features that prevented the detection of the target in the first place, all of them were able to provide a proper explanation.

**How to measure dependence with activation tensor:** To quantitatively check how dependent the explanation maps of CAM-based methods are on the activation tensor, we measured the cosine distance between Activation-CAM and CAM-based methods. However, for natural images like ImageNet, this cosine distance may not give reliable results as the activation tensor may be properly distributed on the target. As in the first row of Figure 5, if the activation tensor is distributed only on the target, it is impossible to distinguish a correct explanation map from an explanation map that depends on the activation tensor. To solve this, we use "Adversarial Example 2 (AE2)" introduced in Sec.2.2. We calculate the cosine distance after making the activation tensor evenly distributed over the four patches, such as the result of Activation-CAM in Figure 6. This approach ensures that the activation tensor is not distributed only to the target, so the dependency can be computed more accurately.(Generating an explanation map similar to Activation-CAM (= Low cosine distance) means it is not capturing the target properly.) We measured the cosine distances for 500 AE2s.

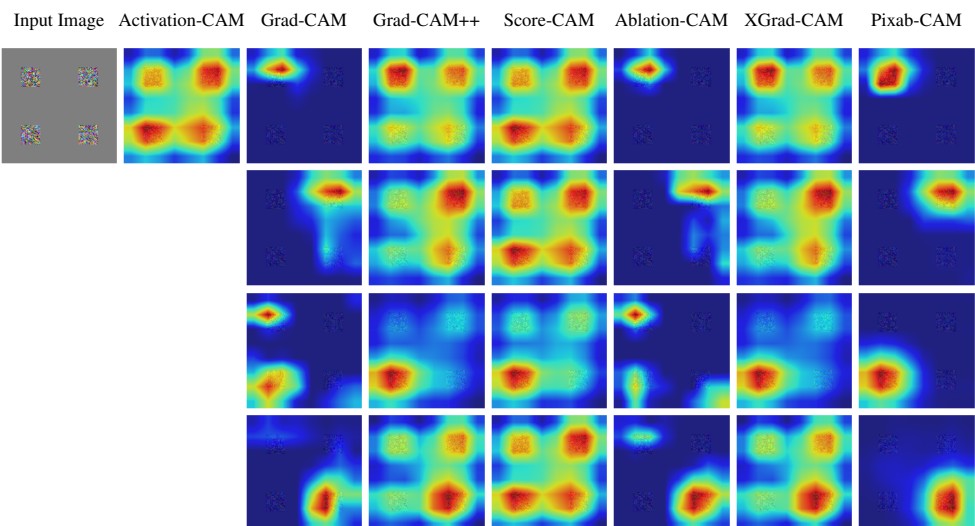

Figure 6: 1st row: "chihuahua", 2nd row: "tusker", 3rd row: "soccer ball", 4th row: "broccoli".

The result for "Dependency on Activation Tensor (DOA)" is in Table 2. Grad-CAM++, Score-CAM, and XGrad-CAM have very low DOA values. In particular, Score-CAM has a value close to 0. In

other words, the explanation map they generate is highly dependent on the activation tensor. We also observed that the less the activation tensor is distributed in the target, the more they depend on the activation tensor. This result implies that in the case of a very weakly activated feature on the image, the explanation maps of these methods may not be able to capture that part even though it is a target. The explanation maps of Grad-CAM and Ablation-CAM do not depend on the activation tensor, but they focus on the wrong place or only part of the target, as shown in Figures 5, 6, and 8.

## 4 PROPOSED METHODOLOGY

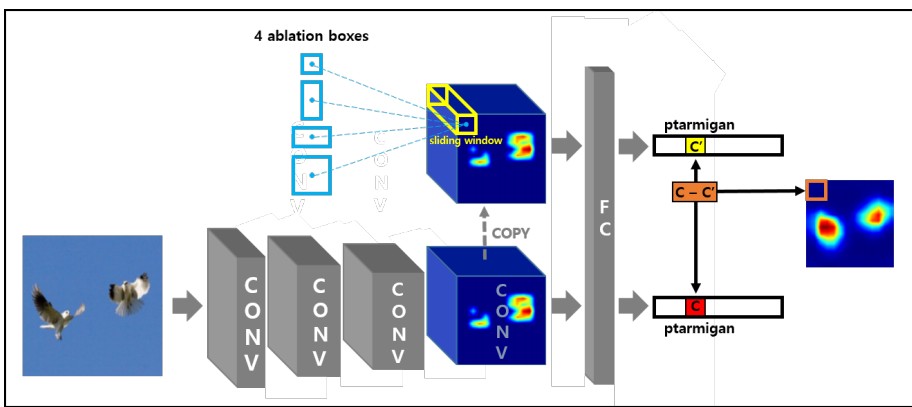

Figure 7: An overview of Pixab-CAM.

Pixab-CAM has three main features. The first is that it doesn't use gradients to avoid problems with gradient saturation that negatively affect visualization quality(Wang et al., 2020; Ramaswamy et al., 2020). Second, we define a pixel-wise weight rather than a channel-wise weight to solve the problem that pixels within a specific channel are tied to a single scalar value. (First motivation in Sec.3.1.) Finally, in order to completely remove the dependency on the activation tensor, we do not include the activation tensor in generating the explanation map. (Second motivation in Sec.3.2.)

As in Eqn.7, we define the weight by using the idea of Ablation-CAM in terms of pixels. Although Ablation-CAM uses only the channel contribution from the deletion(=ablation) perspective when deriving the channel coefficient(Ramaswamy et al., 2020), we define pixel coefficient using both the pixel contribution from the deletion(first term) and preservation(second term) perspectives. If a specific pixel is associated with a target class, the class score will decrease when it is removed(first term) and increase when it appears(second term). These two terms are complementary to each other because they reflect the contribution of pixels from different perspectives.(It is a strict weight assignment method because a high coefficient can be obtained only by showing a high contribution from both perspectives.) As shown in Figure 7, Eqn.7 can be expressed as a tensor composed of 0s(first term) and a tensor composed of 1s(second term), each of size $1 \times 1 \times channel\_size$, moving on the activation tensor in a sliding window manner. We'll call this tensor an ablation box, and use four ablation boxes-$1 \times 1, 1 \times 2, 2 \times 1, 2 \times 2$.(An evaluation of each design choice and why these four ablation boxes were used are covered in Appendix B.) The reason for using multiple scale boxes is to detect objects of various sizes and shapes as in Faster R-CNN(Ren et al., 2015). For reference, even if all boxes are used, the prediction process is only 1/4 of Ablation-CAM. Also, regardless of the size of the box, stride size is 1, and padding is used in all except the $1 \times 1$ box to reflect the pixel contribution fairly. Then, we average over all boxes of each perspective. Finally, the explanation map is obtained after applying ReLU and the normalization function as in Eqn.8. What should be noted here is that the activation tensor is not used in the process of generating the explanation map.

$$p_{ij}^c = \left( \frac{Y^c - Y_{ij}^c}{Y^c} \right) \cdot \left( \frac{\widetilde{Y}_{ij}^c - \widetilde{Y}^c}{\widetilde{Y}^c} \right) \tag{7}$$

$$L_{ij}^c = s \left( ReLU \left( p_{ij}^c \right) \right) \tag{8}$$

where $Y_{ij}^c$ is the score of class $c$ when the pixel at spatial location $(i, j)$ has been replaced by zero, conversely, $\widetilde{Y}_{ij}^c$ is the score of class $c$ when the pixel at spatial location $(i, j)$ is the only remaining pixel and $\widetilde{Y}^c$ is the score of class $c$ when a tensor with all values of 0 is input to the model.

# 5 EXPERIMENTS

In this section, we mainly compare Pixab-CAM with state-of-the-art CAM-based methods. Qualitative (see Sec.5.1) and quantitative evaluations (see Sec.5.2) are performed to evaluate the quality of the explanation map. In addition, we use Inception V3 network(Szegedy et al., 2016) pretrained on the ImageNet provided by TensorFlow as a black box model.

## 5.1 QUALITATIVE EVALUATION (MORE EXAMPLES ARE PROVIDED IN APPENDIX C)

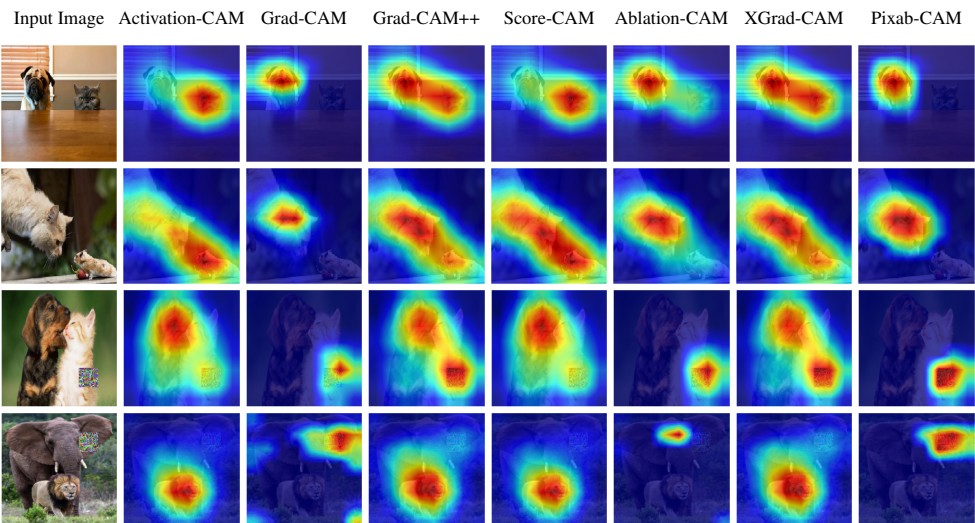

Figure 8: Explanation Map Comparison. (Rows 1-2: Natural Image,   Rows 3-4: AE1)

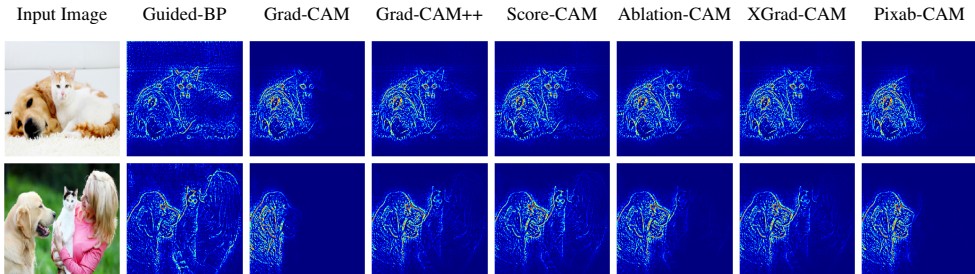

Figure 9: Guided CAM-based Explanation Comparison (Springenberg et al., 2014)

As shown in Figures 8, 9, and 10, we qualitatively compare the explanation maps, guided CAM-based explanations, and weekly supervised localization results generated by Activation-CAM, five state-of-the-art CAM-based methods, and Pixab-CAM. Compared to previous CAM-based methods, Pixab-CAM produces higher quality explanation maps in all examples, whether or not the activation tensor is focusing on the target. On the other hand, the explanation maps of previous CAM-based methods tend not to focus properly on the target. First, Grad-CAM and Ablation-CAM tend to generate insufficient explanation maps, and even Ablation-CAM captures places that are not related to the target class at all. Second, if a part not related to the target class is emphasized in the result of Activation-CAM, that part is also highlighted in the explanation maps of Grad-CAM++, Score-CAM, and XGrad-CAM, which means that they are greatly affected by the activation tensor.

## 5.2 QUANTITATIVE EVALUATION

We scraped 2,000 images with two or more objects with different labels (MOD) to evaluate the performance of the explanation map in situations where fine-grained weight adjustment is required. MOD images consist of objects with different labels but similar textures, such as cats and dogs,

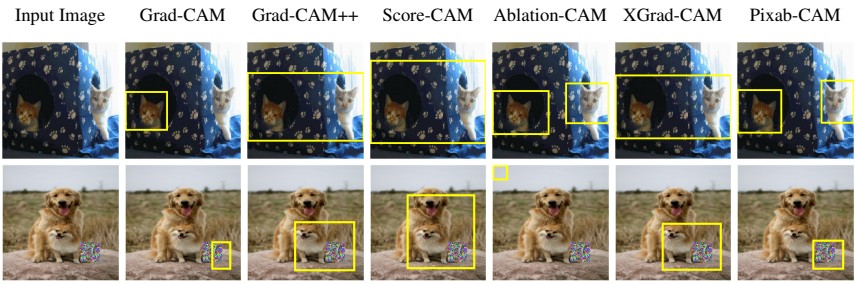

Figure 10: Weakly Supervised Localization Comparison (Oquab et al., 2015)

lions and tigers, trucks and cars, so the explanation map cannot properly capture the target unless the activation tensor is precisely adjusted by weight. In addition, ILSVRC 2012 validation set (IMGN)(Russakovsky et al., 2015), PASCAL VOC 2007 test set (VOC)(Everingham et al., 2010), and COCO 2014 validation set (COCO)(Lin et al., 2014) were used for evaluation.

### 5.2.1 FAITHFULNESS EVALUATION

Table 2: DOA

| Name | DOA |
|---|---|
| Grad-CAM | 0.391 |
| Grad-CAM++ | 0.028 |
| Score-CAM | 0.014 |
| Ablation-CAM | 0.223 |
| XGrad-CAM | 0.031 |
| Pixab-CAM | 0.402 |

Table 3: AD, IIC, and Compactness

| Name | AD (%) | | | | IIC (%) | | | | Compactness | | | |
|---|---|---|---|---|---|---|---|---|---|---|---|---|
| | IMGN | VOC | COCO | MOD | IMGN | VOC | COCO | MOD | IMGN | VOC | COCO | MOD |
| Grad-CAM | 34.8 | 24.1 | 24.5 | 20.0 | 22.0 | 31.3 | 33.1 | 37.6 | 14.7 | 13.7 | 14.2 | 11.3 |
| Grad-CAM++ | 33.7 | 22.4 | 22.8 | 21.3 | 22.1 | 33.3 | 31.0 | 35.2 | 18.4 | 19.8 | 20.4 | 16.9 |
| Score-CAM | 33.2 | 22.8 | 23.6 | 22.8 | 22.2 | 32.5 | 30.8 | 35.0 | 19.4 | 21.1 | 21.7 | 19.4 |
| Ablation-CAM | 32.7 | 21.2 | 22.4 | 20.1 | **24.5** | 33.3 | 32.9 | 37.6 | 16.3 | 17.9 | 18.3 | 14.5 |
| XGrad-CAM | 33.7 | 22.1 | 23.2 | 20.3 | 23.8 | 32.7 | 33.0 | 36.8 | 17.8 | 18.7 | 19.4 | 16.2 |
| Pixab-CAM | **30.2** | **19.4** | **19.1** | **18.9** | 24.2 | **36.2** | **35.9** | **39.7** | 15.4 | 16.1 | 16.6 | 12.6 |

**AD, IIC, and Compactness:** Average Drop (AD)(Chattopadhay et al., 2018) is a metric that measures how much the explanation image, which is generated by point-wise multiplication of the upsampled explanation maps with the original image, maintains the part that plays essential roles in correct decision-making. Increase in Confidence (IIC)(Chattopadhay et al., 2018) measures whether the part that confuses correct decisions in the image has been sufficiently removed. Compactness is the metric proposed for the first time in this paper to measure how large the area the explanation map occupies in an image. An ideal explanation map should be of a suitable size to cover the target compactly. Compactness is expressed as $\frac{1}{N}\sum_{i=1}^{N}(\sum_a \sum_b L^c_{i,(a,b)})$, where $L^c_{i,(a,b)}$ is the explanation map value at spatial location $(a,b)$ for the $i$-th input $X_i$ and $N$ is the number of evaluation images. Results for 1,000 images randomly sampled for each dataset are in Table 3. From the results of IIC and AD, except for IIC in IMGN, Pixab-CAM outperforms other CAM-based methods by large scale on all metrics. This result reveals that Pixab-CAM can successfully leave only the essential parts for the correct decision while sufficiently removing the parts that interfere with decision making. In addition, Pixab-CAM has the second smallest Compactness value after Grad-CAM. It means that Grad-CAM generates an explanation map with a size less than necessary, and other previous CAM-based methods generate an explanation map with excessive size. Meanwhile, Pixab-CAM generates an explanation map of an appropriate size enough to cover the target compactly.

**Deletion and Insertion:** The deletion metric(Petsiuk et al., 2018) measures the decrease in the probability of the predicted class as more and more important pixels are removed. The insertion metric(Petsiuk et al., 2018), on the other hand, measures the increase in probability as more and more pixels are introduced. Table 4 reports the average result of 1,000 images randomly selected from ImageNet. Pixab-CAM exhibited the highest insertion AUC and lowest deletion AUC. This experiment shows that Pixab-CAM successfully assigned coefficients in order of importance.

### 5.2.2 OBJECT LOCALIZATION EVALUATION

In this section, we compare the localization ability of the generated explanation maps. The "Object Localization (OL)" metric was discussed in Sec.2.3 and is therefore omitted from this section. The values in Table 5 are averaged for 1,000 images randomly selected from ImageNet. Pixab-CAM has the highest proportion compared to other CAM-based methods. This means that the explanation maps generated by Pixab-CAM compactly focus the essential parts of the image.

### 5.2.3 "DETECTING ADVERSARIAL PATCH 1" & "DETECTING ADVERSARIAL PATCH 2"

We also evaluate the performance of explanation maps for tasks "Detecting Adversarial Patch 1 (DAP1)" and "Detecting Adversarial Patch 2 (DAP2)", which are proposed for the first time in this paper (see Section 2). The number of patches in the image is randomly selected from 1-9, and the width and height of the patches are also randomly selected from 15%-20% of the input image size. DAP1 and DAP2 were tested on 500 AE1s and 500 AE2s, respectively, and the results are in Table 6. We observe that Pixab-CAM outperforms previous CAM-based methods in both DAP1 and DAP2.

Table 4: Deletion & Insertion

| Name | Deletion | Insertion |
|------|----------|-----------|
| Grad-CAM | 0.090 | 0.433 |
| Grad-CAM++ | 0.094 | 0.428 |
| Score-CAM | 0.101 | 0.425 |
| Ablation-CAM | 0.088 | 0.457 |
| XGrad-CAM | 0.093 | 0.434 |
| Pixab-CAM | **0.082** | **0.469** |

Table 5: Object Localization

| Name | Proportion (%) |
|------|----------------|
| Grad-CAM | 46.39 |
| Grad-CAM++ | 48.17 |
| Score-CAM | 50.77 |
| Ablation-CAM | 51.32 |
| XGrad-CAM | 47.19 |
| Pixab-CAM | **53.22** |

Table 6: DAP1 & DAP2

| Name | DAP1 PG | DAP1 OL | DAP2 PG | DAP2 OL |
|------|---------|---------|---------|---------|
| Grad-CAM | 46.5 | 42.7 | 45.3 | 43.1 |
| Grad-CAM++ | 44.3 | 40.7 | 43.8 | 40.2 |
| Score-CAM | 43.2 | 40.2 | 42.9 | 39.5 |
| Ablation-CAM | 46.8 | 43.1 | 46.7 | 44.9 |
| XGrad-CAM | 44.0 | 41.9 | 43.1 | 40.6 |
| Pixab-CAM | **55.3** | **50.1** | **54.9** | **51.3** |

## 5.3 SANITY CHECK (ADEBAYO ET AL., 2018)

Model parameter randomization test was proposed to compare the explanation map on a trained model to the explanation map on a randomly initialized, untrained network of the same architecture. As in Figure 11, Pixab-CAM is sensitive to model parameters and can judge the quality of the model.

| Input Image | Result | Logit | Mixed_7b | Mixed_6e | Mixed_6c | Mixed_6a | Mixed_5c |
|---|---|---|---|---|---|---|---|

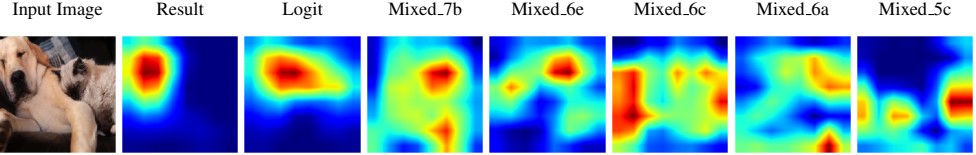

Figure 11: Cascading randomization from top to bottom layers on Inception v3 trained on ImageNet

## 5.4 APPLICATIONS

To prove the applicability of Pixab-CAM in downstream tasks, we applied it to the Brain Tumor Detection task. Since the Brain Tumor dataset is small (N: 253), we used "Transfer Learning"(Pan & Yang, 2009). We downloaded VGG-16(Simonyan & Zisserman, 2014) pretrained on the ImageNet dataset, then trained only the final fully connected layer while keeping the weights of all ConvNets fixed. Figure 12 shows the explanation map on a trained VGG-16 for brain tumor images. Pixab-CAM detects tumor better than other CAM-based methods. This experiment demonstrates that Pixab-CAM can also be applied to fine-tuned models using small datasets.

| Input Image | Grad-CAM | Grad-CAM++ | Score-CAM | Ablation-CAM | XGrad-CAM | Pixab-CAM |
|---|---|---|---|---|---|---|

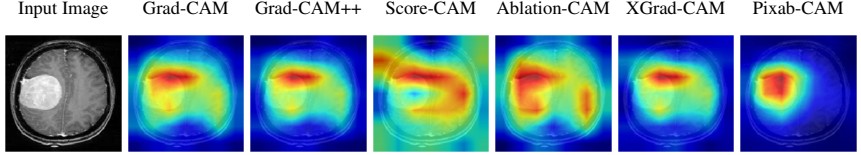

Figure 12: Explanation Map of Brain Tumor images

## 6 CONCLUSION

We proposed Pixab-CAM, a novel CAM-based approach with three characteristics. First, we use pixel-wise weights rather than channel-wise weights so that the importance of the pixels is not distorted. Second, we exclude the activation tensor to create an explanation map independent of the activation tensor. Third, to avoid problems with gradient-based CAMs, gradients are not used when obtaining pixel-wise weights. In addition, we propose a new evaluation metric using adversarial attack and demonstrate the superiority of Pixab-CAM through qualitative and quantitative evaluation.

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

## A    CHANNELS WITH OVERESTIMATED PIXELS

As explained in Sec.3.1, overestimated pixel occurs because pixels within a particular channel share a single value. We visualize channels with overestimated pixels(COP) through Adversarial Example 1 (AE1), Adversarial Example 2 (AE2), and natural image. (Descriptions of AE1 and AE2 are in Sec.2.1 and Sec.2.2, respectively.) For each of Grad-CAM, Grad-CAM++, Score-CAM, Ablation-CAM, and XGrad-CAM, we find the top 20 channels by weight among the 2048 channels in the activation tensor(Inception V3 has 2048 channels(Szegedy et al., 2016)), and then select 6 channels focusing on both the target and non-target part. The visualization results are in Figures 13, 14, and 15. The target in Figures 13, 14, and 15 are the patch, the patch in the upper left, and the "pug" and "labrador retriever", respectively. (The first row, second row, third row, fourth row, and fifth row are the results of Grad-CAM, Grad-CAM++, Score-CAM, Ablation-CAM, and XGrad-CAM respectively, and each column represents a channel.) This result shows that more channels with overestimated pixels(COP) than expected are in the activation tensor.

## B    DESIGN CHOICE

We will find the optimal combination of ablation boxes with the evaluation tasks (DAP1 and DAP2) presented in Sec.2.3. Since it is difficult to evaluate all possible ablation box combinations(For Inception V3(Szegedy et al., 2016), since the size of the activation tensor is $8 \times 8$, the possible ablation box sizes are $1 \times 1, 1 \times 2, 2 \times 1, 2 \times 2, ..., 8 \times 8$), we first reduce the design space through performance evaluation for individual ablation boxes. Tables 7, 8, and 9 are the results when only the first term of Eqn.7 is used, when only the second term is used, and when both the first and second terms are used, respectively. Two characteristics can be found in these tables. First, if any of the width or height of the ablation box is 3, the performance drops significantly. (We also experimented with cases where the ablation box had a width or height of 4 or more, but we did not include it in the table because performance continued to deteriorate.) Second, when the width and height of the ablation box are both less than or equal to 2, the performance of using both the first and second terms of Eqn.7 is always better than the performance of using only the first or second term. It means that the first term and second term are complementary to each other. Based on these results, only ablation boxes of sizes $1 \times 1, 1 \times 2, 2 \times 1$, and $2 \times 2$ are used in the following experiment to find the optimal combination.

Table 10 shows the evaluation results for all possible combinations of the four ablation boxes selected in the previous experiment. (We evaluated all 15 combinations, but combinations that did not include an ablation box of size $1 \times 1$ were not recorded in Table 10 because their performances were all low rank.) This result shows that the performance gets better as the number of ablation boxes increases. For these reasons, we use all four ablation boxes.

## C    MORE QUALITATIVE EVALUATION

Figures 16, 17, 18, 19, and 20 contain more examples of the qualitative evaluation covered in Sec.5.1. (Figure 16: Explanation Map Comparison (Natural Image), Figure 17: Explanation Map Comparison (AE1), Figure 18: Explanation Map Comparison (AE2), Figure 19: Guided CAM-based Explanation Comparison, Figure 20: Weakly Supervised Localization Comparison)

Table 7: Comparison of each ablation box I ("Deletion" Perspective)

| $(1\times1)^-$ | $(1\times2)^-$ | $(2\times1)^-$ | $(2\times2)^-$ | $(1\times3)^-$ | $(3\times1)^-$ | $(2\times3)^-$ | $(3\times2)^-$ | $(3\times3)^-$ | DAP1 PG | OL | DAP2 PG | OL |
|---|---|---|---|---|---|---|---|---|---|---|---|---|
| ✓ | | | | | | | | | 52.6 | **44.3** | 51.9 | **44.8** |
| | ✓ | | | | | | | | 52.4 | 43.9 | **52.3** | 44.6 |
| | | ✓ | | | | | | | 51.9 | 43.8 | 52.0 | 44.1 |
| | | | ✓ | | | | | | **52.8** | 44.1 | 52.2 | 44.4 |
| | | | | ✓ | | | | | 49.8 | 41.7 | 50.3 | 41.7 |
| | | | | | ✓ | | | | 50.1 | 41.4 | 49.9 | 40.9 |
| | | | | | | ✓ | | | 49.4 | 40.9 | 49.7 | 41.2 |
| | | | | | | | ✓ | | 48.8 | 41.0 | 48.8 | 40.7 |
| | | | | | | | | ✓ | 48.5 | 40.6 | 48.9 | 40.9 |

*Combinations of Ablation Boxes (Superscript $-$ means the first term of Eqn.7)*

Table 8: Comparison of each ablation box II ("Preservation" Perspective)

| $(1\times1)^+$ | $(1\times2)^+$ | $(2\times1)^+$ | $(2\times2)^+$ | $(1\times3)^+$ | $(3\times1)^+$ | $(2\times3)^+$ | $(3\times2)^+$ | $(3\times3)^+$ | DAP1 PG | OL | DAP2 PG | OL |
|---|---|---|---|---|---|---|---|---|---|---|---|---|
| ✓ | | | | | | | | | **52.9** | 44.0 | **52.2** | 44.1 |
| | ✓ | | | | | | | | 52.2 | 43.8 | 52.0 | 44.2 |
| | | ✓ | | | | | | | 51.7 | **44.3** | 51.9 | 43.9 |
| | | | ✓ | | | | | | 52.5 | 43.8 | 51.3 | **44.3** |
| | | | | ✓ | | | | | 48.9 | 41.4 | 50.1 | 41.8 |
| | | | | | ✓ | | | | 49.9 | 41.0 | 49.5 | 41.1 |
| | | | | | | ✓ | | | 49.5 | 40.3 | 49.7 | 41.0 |
| | | | | | | | ✓ | | 48.5 | 41.3 | 47.8 | 41.2 |
| | | | | | | | | ✓ | 48.5 | 40.9 | 48.3 | 41.4 |

*Combinations of Ablation Boxes (Superscript $+$ means the second term of Eqn.7)*

Table 9: Comparison of each ablation box III ("Deletion + Preservation" Perspective)

| $(1\times1)^\mp$ | $(1\times2)^\mp$ | $(2\times1)^\mp$ | $(2\times2)^\mp$ | $(1\times3)^\mp$ | $(3\times1)^\mp$ | $(2\times3)^\mp$ | $(3\times2)^\mp$ | $(3\times3)^\mp$ | DAP1 PG | OL | DAP2 PG | OL |
|---|---|---|---|---|---|---|---|---|---|---|---|---|
| ✓ | | | | | | | | | **53.8** | 47.1 | **53.3** | **46.6** |
| | ✓ | | | | | | | | 52.8 | 46.5 | 52.3 | 45.9 |
| | | ✓ | | | | | | | **53.8** | 46.4 | 52.7 | 45.5 |
| | | | ✓ | | | | | | 53.5 | **48.1** | 53.1 | 45.7 |
| | | | | ✓ | | | | | 48.4 | 41.1 | 50.1 | 40.7 |
| | | | | | ✓ | | | | 50.2 | 41.3 | 49.1 | 40.5 |
| | | | | | | ✓ | | | 50.2 | 41.1 | 48.9 | 41.3 |
| | | | | | | | ✓ | | 48.4 | 41.6 | 47.8 | 41.0 |
| | | | | | | | | ✓ | 47.8 | 41.8 | 48.1 | 40.6 |

*Combinations of Ablation Boxes (Superscript $\mp$ means the first and second terms of Eqn.7)*

Table 10: Find the optimal combination of ablation boxes

| $(1\times1)^\mp$ | $(1\times2)^\mp$ | $(2\times1)^\mp$ | $(2\times2)^\mp$ | DAP1 PG | OL | DAP2 PG | OL |
|---|---|---|---|---|---|---|---|
| ✓ | | | | 53.8 | 47.1 | 53.3 | 46.6 |
| ✓ | ✓ | | | 53.8 | 47.5 | 52.9 | 47.7 |
| ✓ | | ✓ | | 54.1 | 47.5 | 53.3 | 48.0 |
| ✓ | | | ✓ | 54.4 | 47.8 | 53.1 | 48.4 |
| ✓ | ✓ | ✓ | | 54.6 | 48.4 | 53.6 | 49.3 |
| ✓ | ✓ | | ✓ | 55.1 | 49.4 | 54.5 | 51.0 |
| ✓ | | ✓ | ✓ | 54.9 | 49.9 | 54.2 | 50.6 |
| ✓ | ✓ | ✓ | ✓ | **55.3** | **50.1** | **54.9** | **51.3** |

*Combinations of Ablation Boxes*

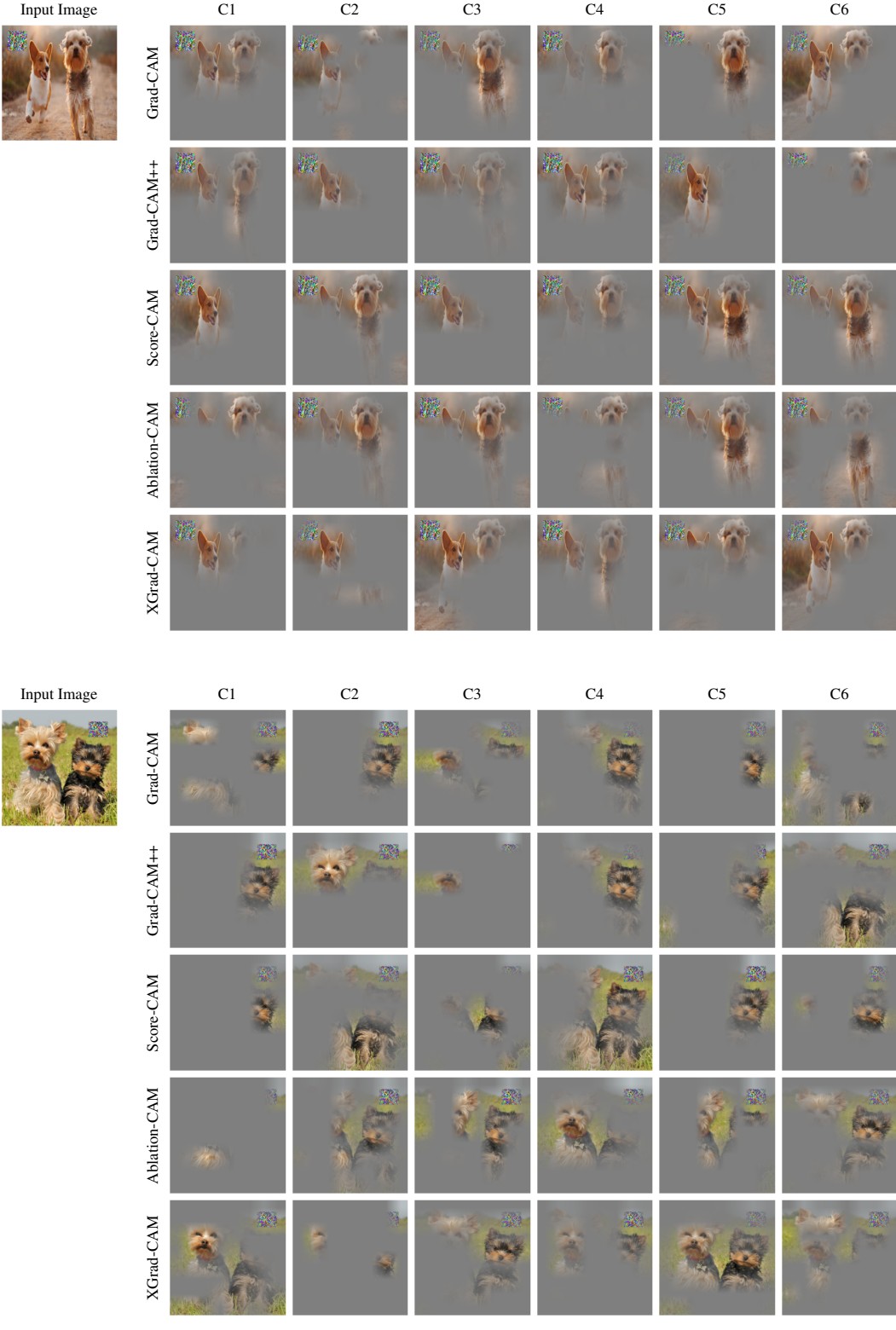

Figure 13: Adversarial Example 1

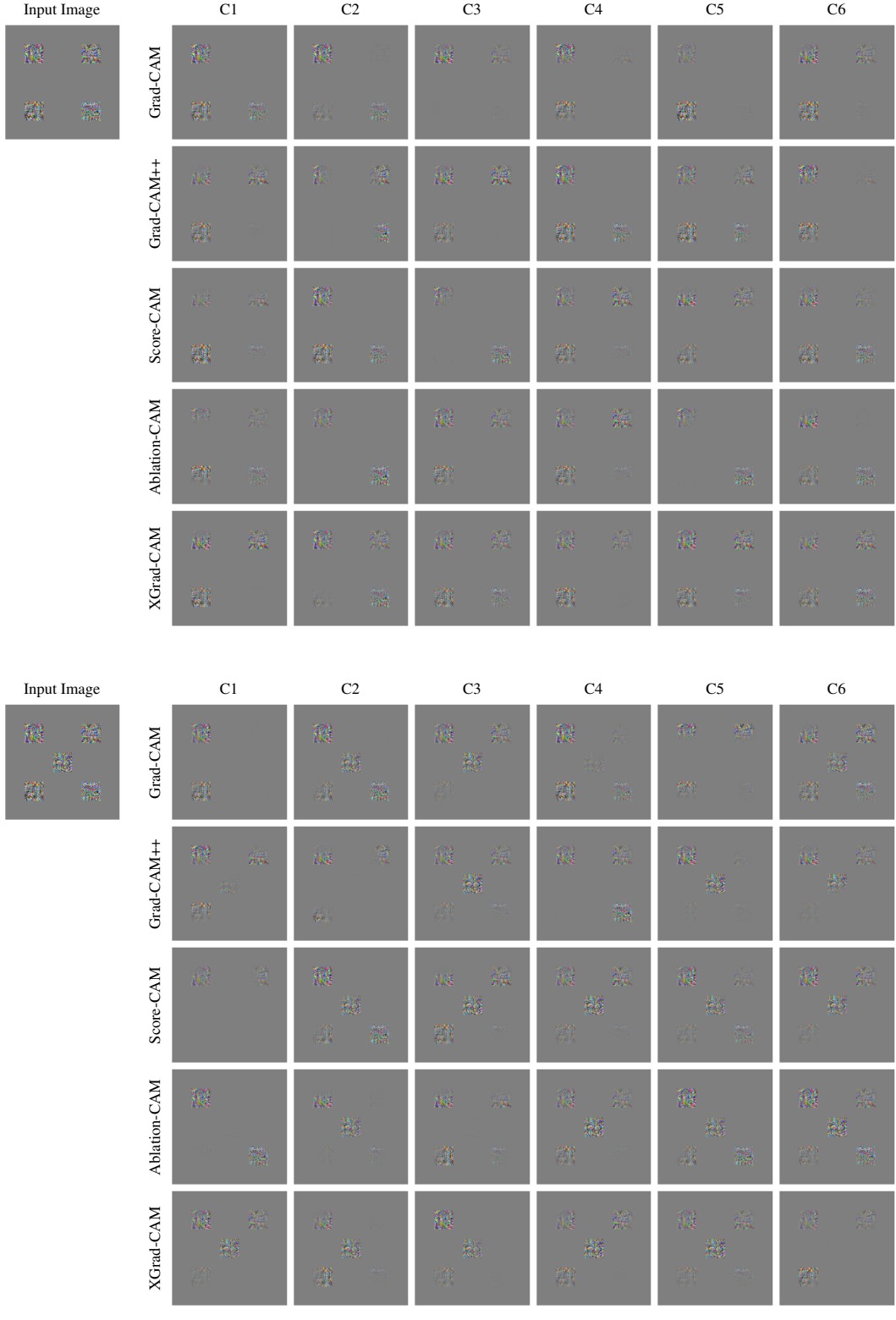

Figure 14: Adversarial Example 2

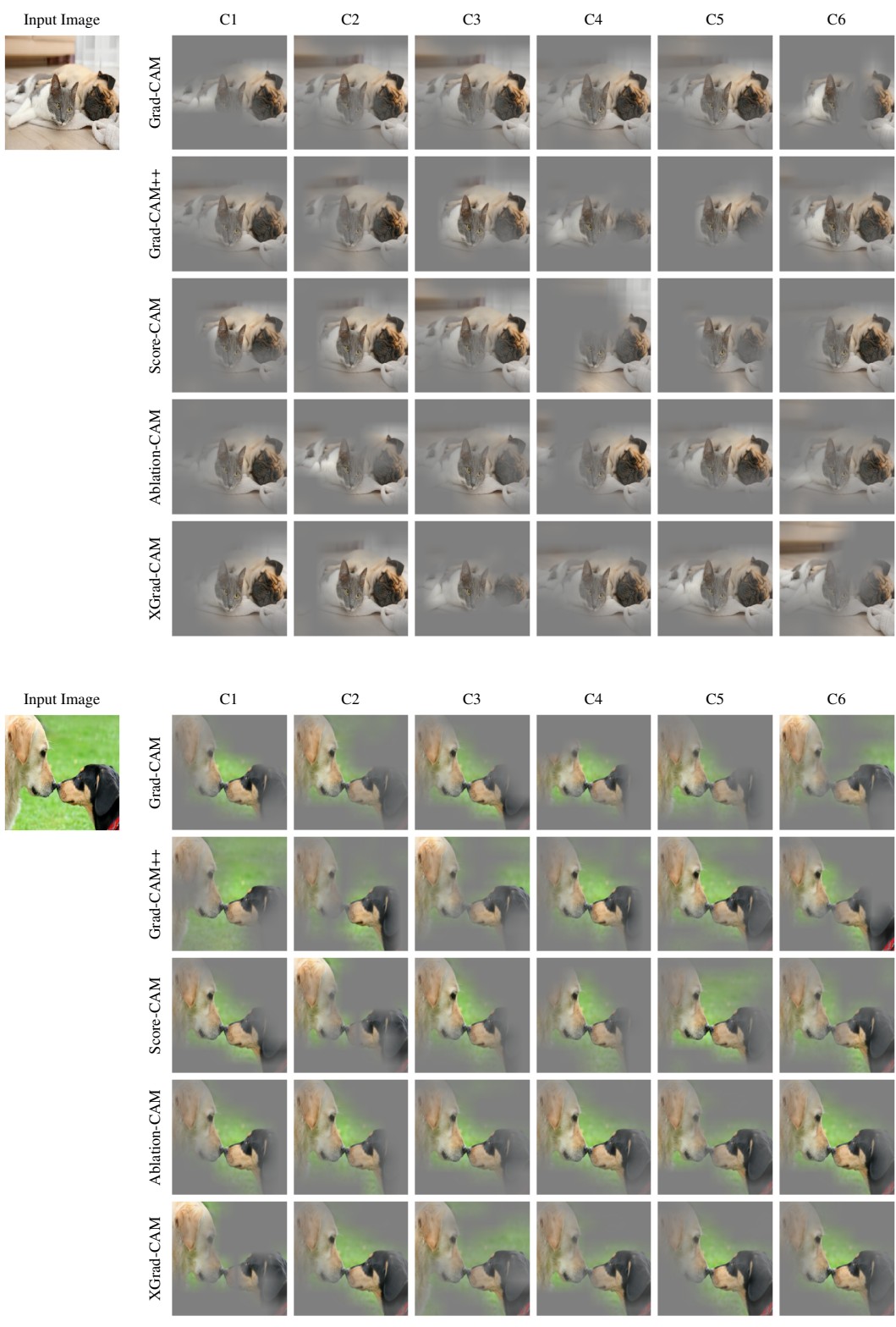

Figure 15: Natural Image

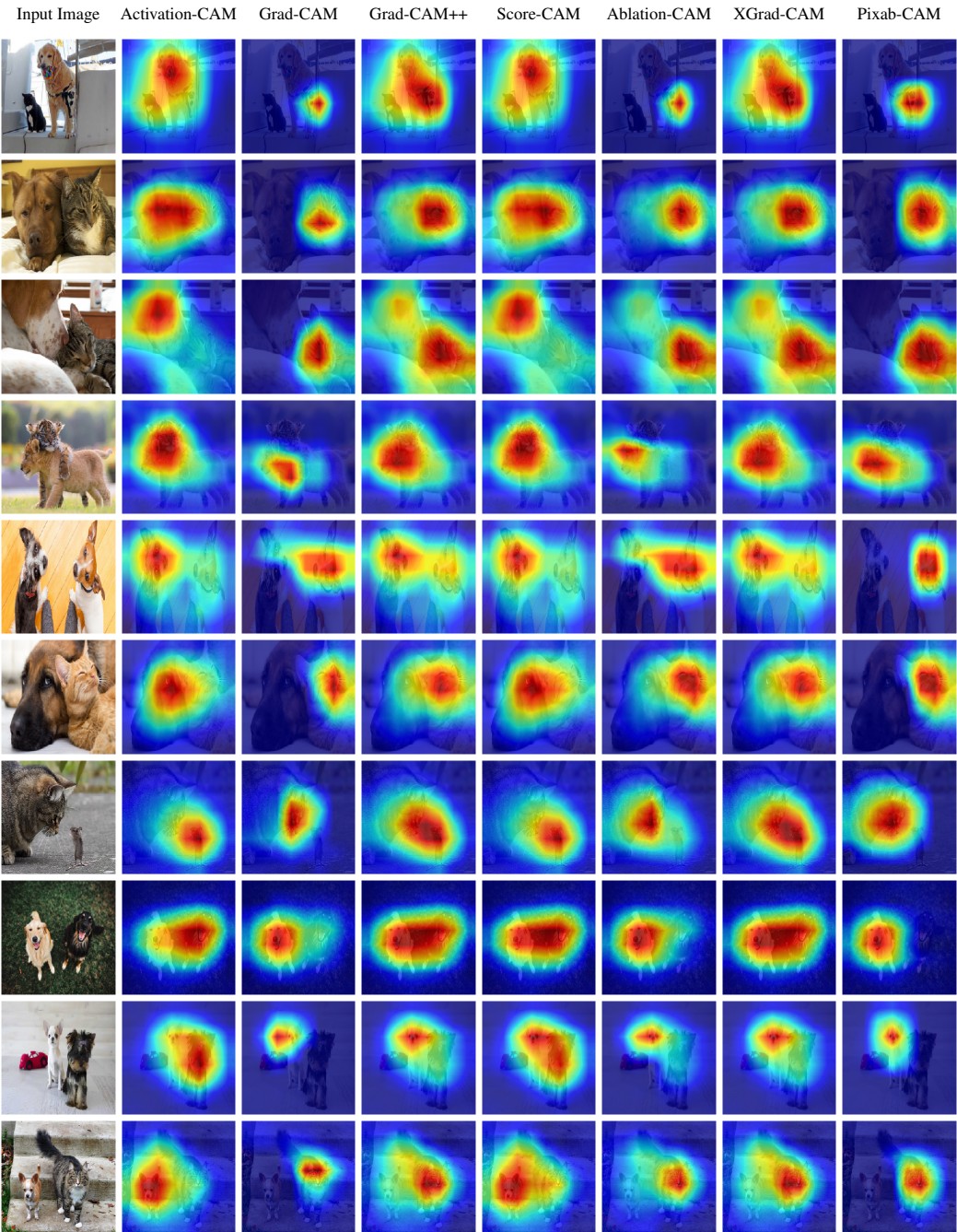

Figure 16: Explanation Map Comparison (Natural Image)

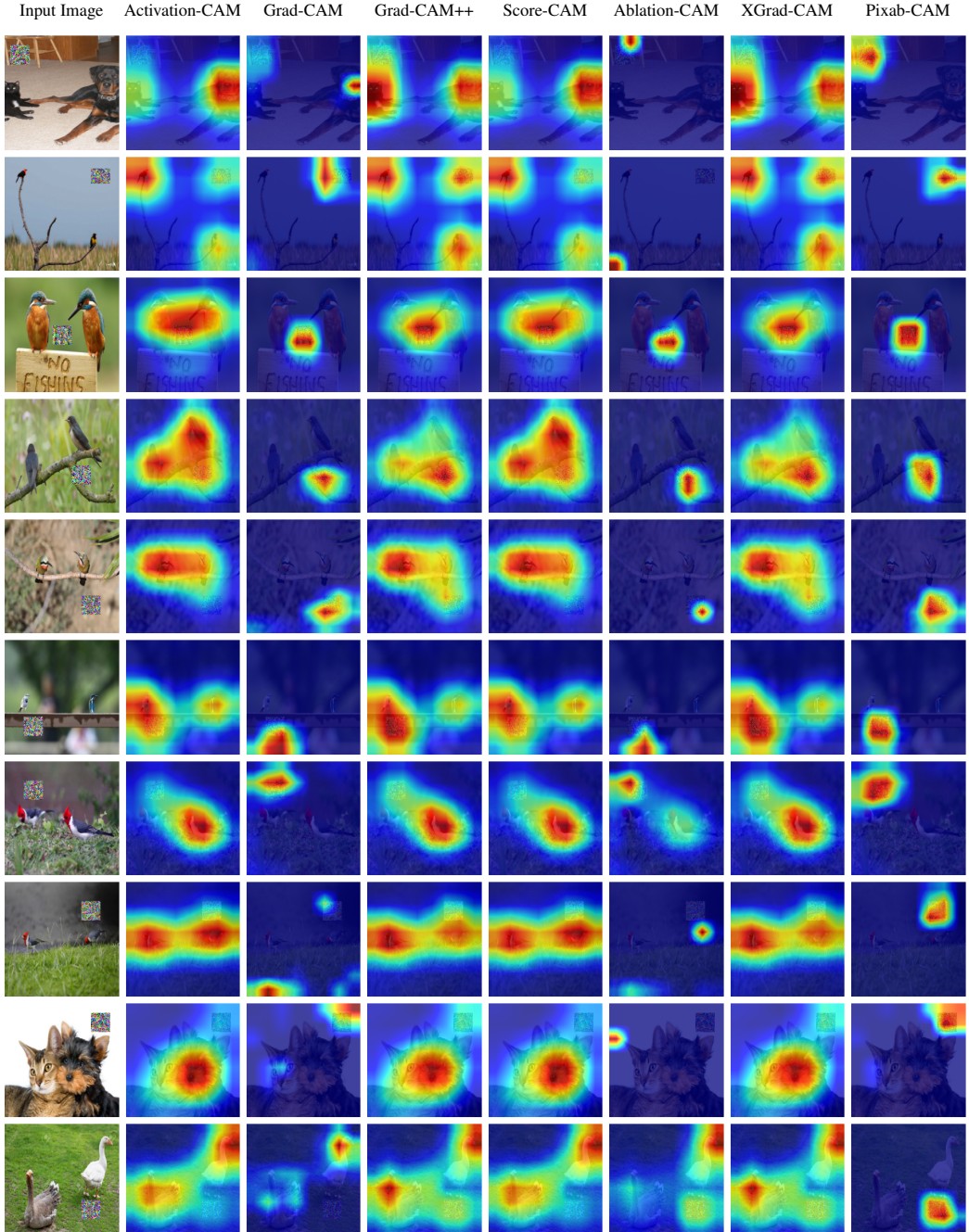

Figure 17: Explanation Map Comparison (AE1)

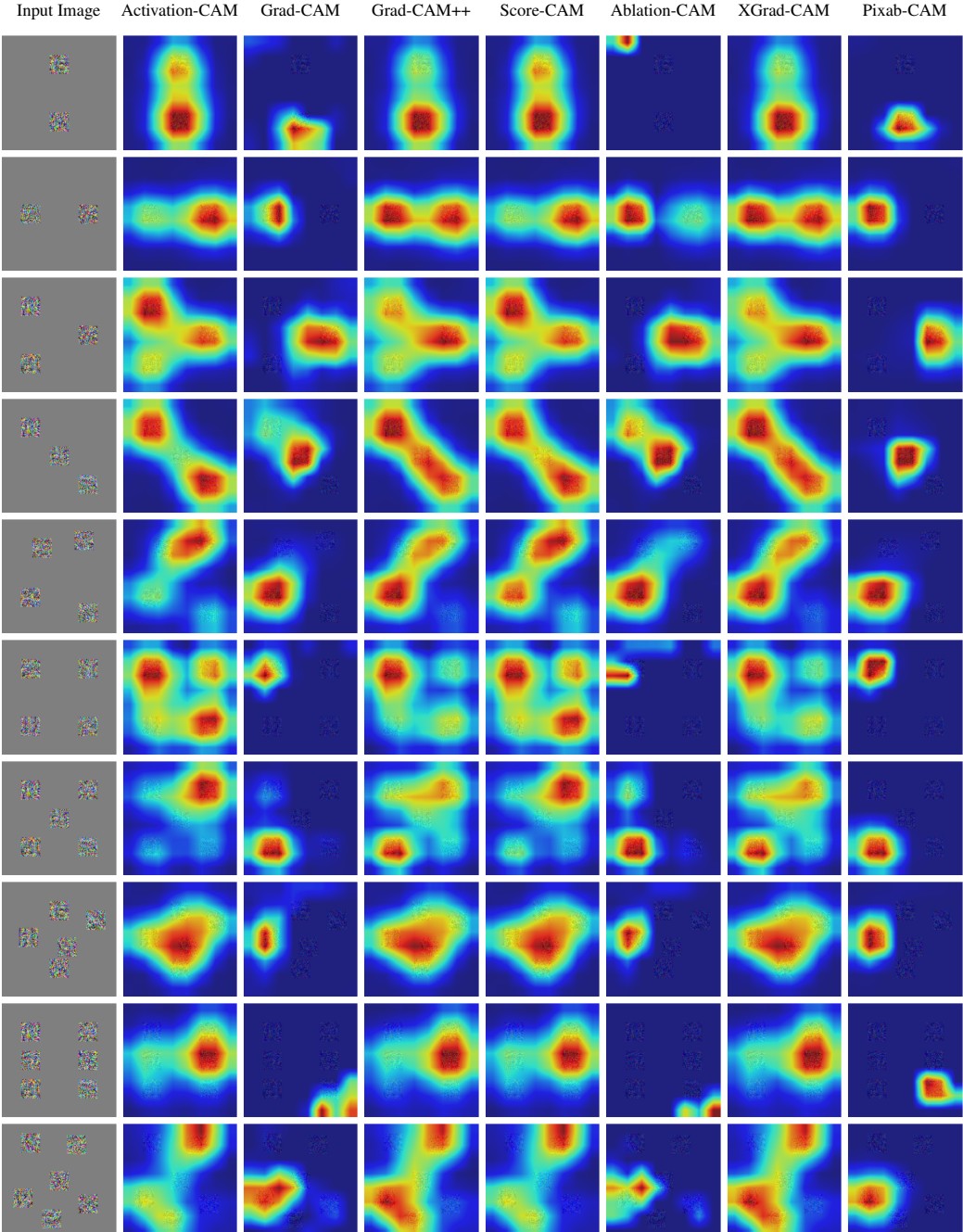

Figure 18: Explanation Map Comparison (AE2)

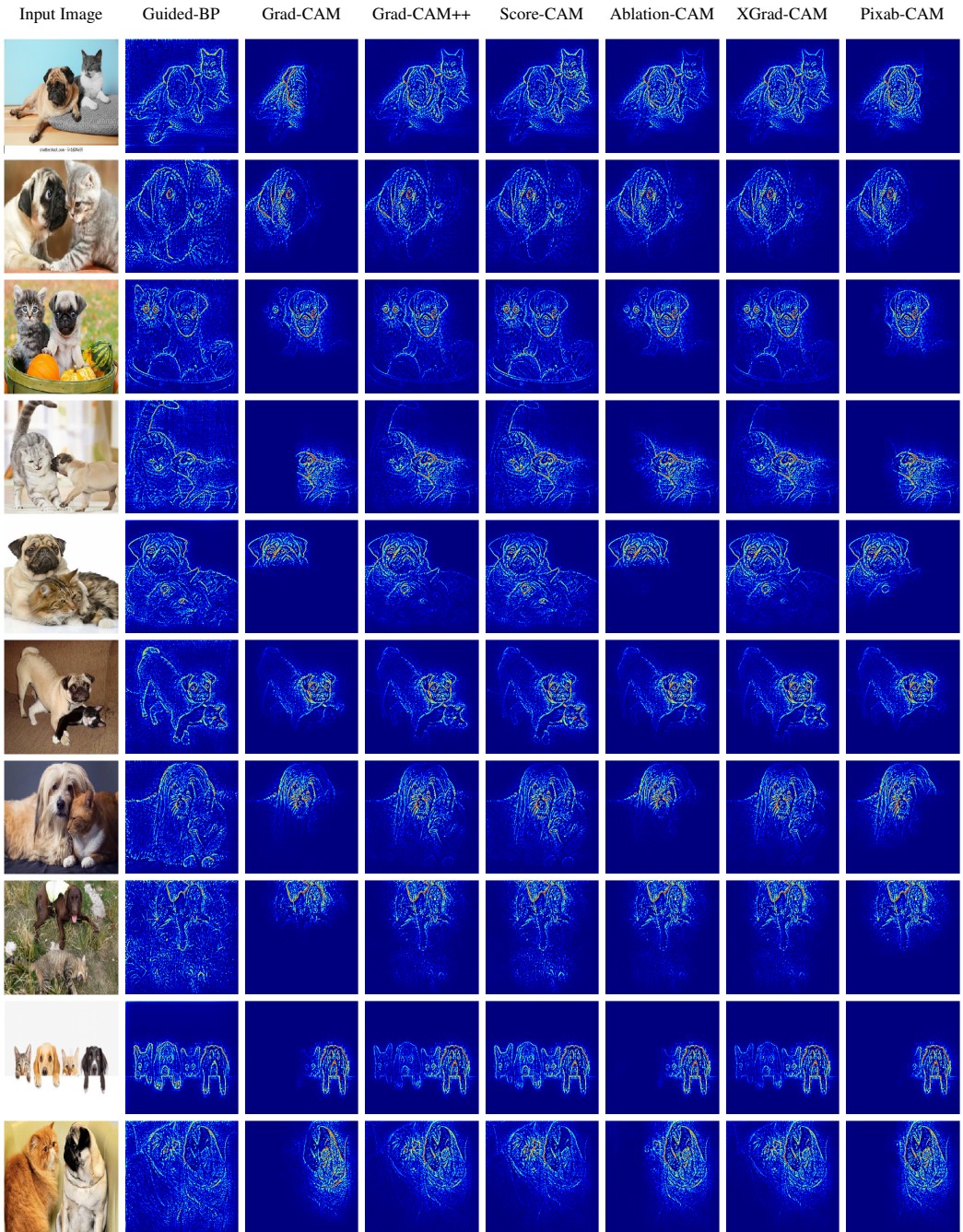

Figure 19: Guided CAM-based Explanation Comparison

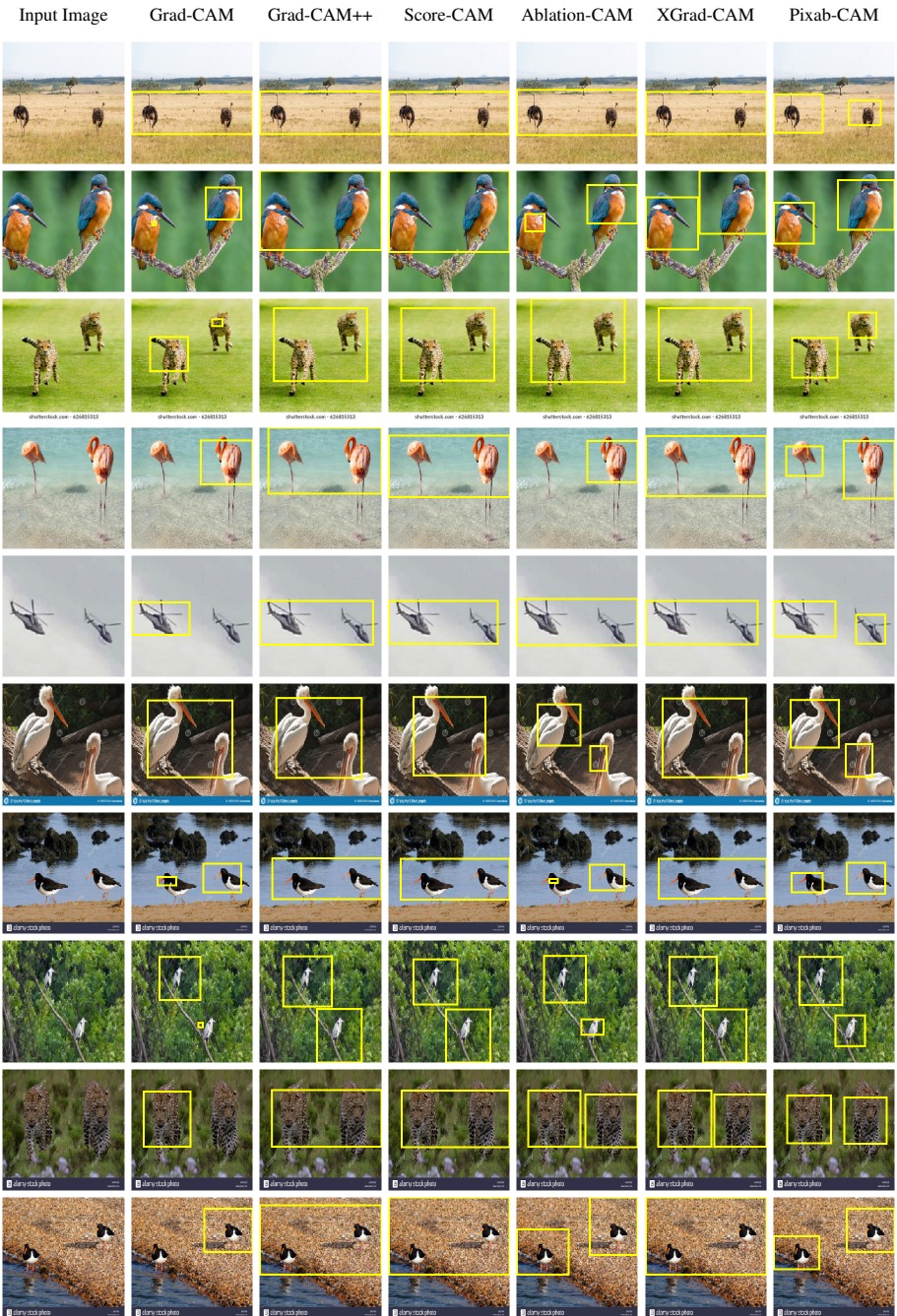

Figure 20: Weakly Supervised Localization Comparison

