# OpenReview forum: "Pixab-CAM: Attend Pixel, not Channel"
_ICLR.cc/2022/Conference — ICLR 2022 Submitted_

### Official Review · Reviewer_8WRU · 2021-11-01

**Correctness:** 2
**Technical Novelty And Significance:** 2
**Empirical Novelty And Significance:** 2
**Recommendation:** 3
**Confidence:** 4

**Main Review:**

In this paper, the author proposes a new variant of class activation map (CAM), i.e., Pixab-CAM with a novel evaluation metric to show the effectiveness of the proposed Pixab-CAM.

In general, the Pixab-CAM provides a new view of CAM-related methods, especially on pixel-level attention. However, I have the following serious concerns:

1. The writing should be improved. I see some serious grammar and punctuation mistakes. The author should improve their writing skills.

2. The pixel-level idea is not a brand new idea in the computer vision community. It has been used in the field of image recognition [1], visual explanation [2], and many other areas including adversarial attack. The author should at least cite this paper and discuss the differences between their modules and the proposed Pixab-CAM. Especially, [2] proposed an attention branch, which can also perform visual explanation with spatial attention and boost the performance. The author should definitely discuss with that paper.

3. More recent results and methods should be included in this paper and fewer figures should be included. First, compared to existing works like Grad-CAM, Grad-CAM++ and ScoreCAM, the proposed Pixab-CAM only lists some regular metrics in Table 2 and Table 3. I would suggest the author list all metrics in previous papers in the appendix. Also, there are some recent CAM related methods like [3], I think the author should also include them for comparison, not just for visualization methods.

Based on these three drawbacks, I would like to recommend the author revise this paper for the next conference.

References:

1. Woo S, Park J, Lee J Y, et al. CBAM: Convolutional block attention module[C]//Proceedings of the European conference on computer vision (ECCV). 2018: 3-19.
2. Fukui H, Hirakawa T, Yamashita T, et al. Attention branch network: Learning of attention mechanism for visual explanation[C]//Proceedings of the IEEE/CVF Conference on Computer Vision and Pattern Recognition. 2019: 10705-10714.
3. Bae W, Noh J, Kim G. Rethinking class activation mapping for weakly supervised object localization[C]//European Conference on Computer Vision. Springer, Cham, 2020: 618-634.

**Summary Of The Paper:**

In this paper, the author proposes a new variant of class activation map (CAM), which is a gradient-free and pixel-wise CAM variant and named Pixab-CAM. The author wants to claim that the proposed Pixab-CAM is superior to previous CAM-based methods on normal metrics, and propose a novel evaluation metric using an adversarial attack.

**Summary Of The Review:**

In this paper, the author proposes a new variant of class activation map (CAM), i.e., Pixab-CAM with a novel evaluation metric to show the effectiveness of the proposed Pixab-CAM.

However, several drawbacks including writing, novelty and fair comparison suggests me to reject this paper.

---

### Official Review · Reviewer_u8GW · 2021-11-01

**Correctness:** 4
**Technical Novelty And Significance:** 3
**Empirical Novelty And Significance:** 3
**Recommendation:** 6
**Confidence:** 4

**Main Review:**

Strengths:
+ Easy to read and understand.
+ The motivation is clear and well-illustrated.
+ It is interesting to connect adversarial attack and interpretability.

Weaknesses/Concerns:
- The adversarial example (AE1) is kind of confusing. In Figure 1, (b) is an adversarial patch that is classified to a targeted class, and then the author directly put this patch into the input image (c). It is misleading to call (e) as an adversarial example, as there is no evidence showing that (e) is misclassified by the model (should it be?). Or the adversarial patch just works as disturbance?
- What is Figure (4)? Why does the activation map look like this? Intuitively, should it be a coarse heatmap?
- In Figure (8) rows3-4, why it is expected to highlight the adversarial patches? What is the label class / predicted class of the input image?
- In sec 3.2, to avoid the activation tensor/map properly distributed on a single target, is it also practical to use images with two or more objects instead of adversarial patches? In this case, the activation tensor should also be randomly distributed.

**Summary Of The Paper:**

This paper aims to solve two common limitations in CAM-based methods, and proposes a new type of CAM (Pixab-CAM) that utilize pixel-wise weights instead of channel-wise weight. Meanwhile, the author also proposes to use adversarial attack as a novel evaluation metric. This work demonstrates to be superior to previous CAMs on corresponding metrics.

**Summary Of The Review:**

Overall, I vote for accepting, even with some concerns unsolved.

The pipeline of Pixab-CAM is simple and easy to adopt. This paper also provide comprehensive analysis of its motivation in section 3.2. The adversarial patch is novel and contributes to this filed.

---

### Official Review · Reviewer_roRV · 2021-11-02

**Correctness:** 2
**Technical Novelty And Significance:** 3
**Empirical Novelty And Significance:** 3
**Recommendation:** 5
**Confidence:** 4

**Main Review:**

### Strengths
- The method is simple and intuitive.
- Visualized examples help authors understand.
- Code is provided, so it is likely to be reproducible.

&nbsp;
### Weaknesses
Overall, the experiments were conducted only on a small number of images, so it is questionable whether they can be reliable.

- **Quantitative results**: For all quantitative results, only 1,000 images were used for each dataset. This can be a very small weight depending on the entire dataset, and it can cause a significant selection bias. In other words, depending on how 1,000 images are chosen, the results can be different. The authors argue that the time complexity of the proposed method is not high, then why not use the entire dataset? In particular, this topic does not require a training process, so the experiment would not take much time. At least, it seems necessary to repeat the experiment multiple times and get the confidence interval of the performance.
- **Sanity check & Applications**: Results for two sections only exist for one image. I know it is hard to add them to the body text, but I cannot find them in the supplementary file either.

&nbsp;
### Other Comments
In addition to Weaknesses, there are concerns about the following items.

- **Readability**: The location of tables, formulas, and figures is not suitable for readability. For example, it seems better to arrange Tables 2, 4, and Eq.(7) as close as possible to the position described in the text. And, although it is the contents of the supplementary file, Tables 7~9 are also inefficiently organized. If "Combination of Ablation boxes" are grouped into one column and placed each item in each row, it seems to be better in terms of space utilization and readability.
- **Qualitative results**: When showing an example like Figure 8, please indicate which class the CAM is for. Also, if there are multi-class objects in one image, it would be better to show all CAMs for each class.
- **Time complexity**: There is only mention that the prediction process is 1/4 of Ablation-CAM, but there is no detailed comparison of time complexity. What is the time complexity (FPS) for each CAM?

&nbsp;

**Summary Of The Paper:**

This paper proposes a method for generating an explanation map (Pixab-CAM) showing which part the CNN model for classification refers to in making a decision. Pixab-CAM utilizes how the classification results are changed by focusing or removing a specific region of the convolutional feature map. By using this, the authors claim that various problems (gradient saturation, single scalar value for a specific channel, dependency on the activation tensor) of existing CAM-based methods are solved. Experiments were conducted on various datasets and metrics, and through this, the effectiveness of the proposed method was verified.

**Summary Of The Review:**

The proposed method has the advantage of being intuitive and simple, but it is suspicious whether the experiments to support its effectiveness are reliable. Therefore, my initial rating is 5, and I expect the authors to address my concerns through a rebuttal.

---

### Official Review · Reviewer_Kcad · 2021-11-04

**Correctness:** 2
**Technical Novelty And Significance:** 2
**Empirical Novelty And Significance:** 3
**Recommendation:** 3
**Confidence:** 4

**Main Review:**

Strengths:

Many of the ideas in the paper sound interesting.

Weaknesses:

The paper is not written or structured clearly. The authors make many claims that are not adequately justified. It would be very helpful if they could define with equations the objects they are referring to (for example, they never define the activation tensor). I am not even sure I understood what the method actually does because I am unsure what it means for a pixel to be "the only remaining pixel at a location" or that a zero tensor is input to the model. In general the paper feels incomplete. For example, the sanity check in Section 5.3 consists of a single example.

Regarding the proposed tasks based on adversarial examples, I think it is intriguing, but I again do not think that they are adequately justified. The authors state: "If a targeted adversarial attack is made on a specific patch in the zero image, it is self-evident that CNN sees that patch and determines that it is a target, so the performance can be evaluated by whether the explanation map correctly captures that part." I find this explanation very confusing. It would be great if the authors explain more clearly what the advantage is with respect to using images where we know the location of the relevant object.

Additional comments:

The proposed method is built upon the strategy of checking how the classifier degrades under input ablation. It combines the intuition of Ablation-CAM and Score-CAM and operates at the pixel level, which is an interesting idea. However, as mentioned in [1], such a strategy has an important drawback -- samples where a subset of the features are removed come from a different distribution, which violates one of the key assumptions in machine learning: the training and evaluation data come from the same distribution. The empirical result in [1] implies that strong performance degradation without re-training might be caused by a shift in distribution instead of removal of information. The authors should at least comment on these issues. In addition, I would suggest using benchmark metrics such as PxAP [2].

[1] Hooker, Sara, et al. “A Benchmark for Interpretability Methods in Deep Neural Networks.” Advances in Neural Information Processing Systems 32 (2019): 9737-9748.
[2]Choe, Junsuk, et al. “Evaluating weakly supervised object localization methods right.” Proceedings of the IEEE/CVF Conference on Computer Vision and Pattern Recognition. 2020.

**Summary Of The Paper:**

This paper proposes an ablation-based method to produce class activation maps, as well as two new evaluation procedures based on adversarial examples.

**Summary Of The Review:**

The paper contains some intriguing ideas, but I cannot recommend acceptance due to the lack of clarity and justifications.

---

### Decision · Program_Chairs · 2022-01-20

**Decision:**

Reject

**Comment:**

This work received borderline rates with slight preference to rejection. The main concerns range from writing, novelty to empirical evaluations. Given that no authors’ responses are submitted, we have decided to reject this work.